# Modifying Elder's Longitudinal Dispersion Coefficient for Two-Dimensional Solute Mixing Analysis in Open-Channel Bends

Kyong Oh Baek [1],* and II Won Seo [2]

1   Department of Civil and Environmental Engineering & Construction Engineering Research Institute, Hankyong National University, Anseong 17579, Korea
2   Institute of Construction and Environmental Engineering & Department of Civil and Environmental Engineering, Seoul National University, Seoul 08826, Korea
*   Correspondence: pko@hknu.ac.kr; Tel.: +82-31-670-5141

**Abstract:** Elder's equation for the longitudinal dispersion coefficient in two-dimensional solute transport analysis cannot be applied to curved channels because the vertical distribution of the longitudinal velocity does not obey the logarithmic law in the bends of an open channel. In this study, a two-dimensional longitudinal dispersion coefficient based on an equation that can appropriately describe the vertical distribution of flow velocity in open-channel bends is derived theoretically. The proposed equations for the vertical velocity distribution and dispersion coefficient are compared and verified with values measured from two different types of open channels, i.e., a laboratory channel and a natural-like channel. The increase in the longitudinal dispersion coefficient based on the difference in the vertical distribution of the flow velocity is evaluated quantitatively. In terms of the longitudinal dispersion coefficient, no significant difference is observed between the observed dispersion coefficient based on the concentration data and the coefficient value calculated using the equation proposed in this study. The dispersion equation proposed in this study can be easily applied to assign the value of the longitudinal dispersion coefficient for the two-dimensional mixing modelling in bends using basic hydraulic factors.

**Keywords:** two-dimensional mixing; vertical distribution of velocity; longitudinal dispersion coefficient; curved channel; hydraulic factor

## 1. Introduction

Herein, we discuss a theoretical method for estimating the longitudinal dispersion coefficient for solute transport in an open-channel flow. It is noteworthy that the longitudinal dispersion coefficient in a one-dimensional analysis and that in a two-dimensional analysis are not identical. The value of the two-dimensional longitudinal dispersion coefficient is much lower than that of the one-dimensional dispersion coefficient. The former considers only the vertical distribution of longitudinal shear flow, whereas the latter accounts for both the vertical and horizontal distributions of shear flow [1–4]. There have been numerous studies on the one-dimensional dispersion coefficient since 2000, such as [5–22]. Surprisingly, there are few studies for the two-dimensional longitudinal dispersion coefficient compared to studies for the one-dimensional coefficient [23,24].

For two-dimensional solute transport in an open-channel flow, for the first time, Elder [25] theoretically derived the longitudinal dispersion coefficient, assuming the vertical distribution of the longitudinal velocity as the logarithmic function proposed by van Karman [26] in the infinitely wide-open channel:

$$u - \overline{u} = \frac{u_*}{\kappa}\left(1 + \ln y'\right) \tag{1}$$

where $u$ is the longitudinal velocity, $\overline{u}$ is the vertically averaged velocity, $u_*$ is the frictional velocity, $\kappa$ is the von Karman constant, $y'$ is the dimensionless vertical coordinate defined as $y/d$, and $d$ is the water depth. The longitudinal dispersion coefficient can be derived using the triple integral [1,25,27]:

$$D_L = -\frac{1}{d} \int_0^d u' \int_0^y \frac{1}{\varepsilon} \int_0^y u' dy dy dy \tag{2}$$

where $D_L$ is the longitudinal dispersion coefficient; $u'(= u - \overline{u})$ is the velocity deviation; and $\varepsilon$ is the vertical diffusion coefficient, which has a vertical distribution as follows [26]:

$$\varepsilon = \kappa u_* y \left(1 - y'\right) \tag{3}$$

By substituting Equations (1) and (3) into Equation (2), the triple integral result is:

$$D_L = \frac{0.404}{\kappa^3} du_* \tag{4}$$

Elder [25] proposed a longitudinal dispersion coefficient with a van Karman constant ($\kappa$) of 0.41 and added the depth-averaged value for Equation (3) ($\overline{\varepsilon} = 0.067\ du_*$), as follows:

$$D_L = (5.86 + 0.067)du_* = 5.93du_* \tag{5}$$

This expression has a theoretical background and is expressed using simple constants, as shown in Equation (5); hence, it has been widely used to determine the longitudinal dispersion coefficient in two-dimensional solute transport analysis.

However, Elder's equation (Equation (5)) is inadequate for application to curved channels because the vertical distribution of the longitudinal velocity does not obey the logarithmic law in the bends of an open channel. The vertical distribution of longitudinal velocity in the bends is shown in Figure 1. As shown in this figure, the maximum velocity occurs near or below the center of the water depth [28]. According to Blankaert [29], Blankaert and de Vriend [30], and Baek and Seo [31], the vertical distribution is more similar to a parabolic function instead of a logarithmic one. Hence, an equation that can describe the vertical velocity distribution of open-channel bends appropriately is to be proposed.

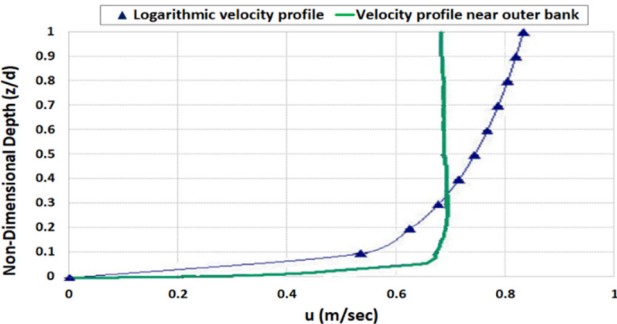

**Figure 1.** Comparison of vertical distribution for longitudinal flow velocity in straight and curved channels (adapted from Mozaffari et al. [28]).

In this study, we theoretically derived a two-dimensional longitudinal dispersion coefficient based on an equation that can appropriately describe the vertical distribution of flow velocity in open-channel bends. The proposed equations for the vertical velocity distribution and dispersion coefficient were compared and verified with values measured from two different types of open channels, i.e., a small-scale laboratory channel and a mid-scale natural-like channel. The increase in the longitudinal dispersion coefficient based on the difference in the vertical distribution of the flow velocity was evaluated quantitatively.

## 2. Materials and Methods

### 2.1. Velocity Profile in Bends

Owing to the channel curvature of natural rivers, the vertical distribution of flow velocity does not always obey a logarithmic distribution; therefore, Elder's equation tends to underestimate the dispersion coefficient. In particular, the flow velocity distribution becomes distorted in bends or reaches with a significant amount of vegetation [29,32–34].

In this case, the longitudinal dispersion coefficient can be calculated via the vertical distribution of the flow velocity instead of the logarithmic distribution. Patil and Singh [33] calculated the dispersion coefficient based on different vegetation densities using the power law for the velocity distribution equation. Their method is advantageous in that the velocity distribution and dispersion coefficient values can be calculated variously based on the change in power. For a curved open channel, Mozaffari et al. [28] proposed an equation that adds a sine function to a power function, as follows:

$$u - \overline{u} = \frac{Au_*}{\kappa}(y' - 0.1)^{0.5} + B sin^2 \pi y' \tag{6}$$

where $A$ and $B$ are regression coefficients that must be determined from experimental data. Equation (6) is based on the power function form, as shown in Figure 2. By adding a sine function form that exhibits the maximum value at the center, the convex flow velocity distribution equation can be reproduced in the central region of the water depth. In addition, because the sine function term can be regarded as an additional momentum term caused by curvature, if constant B is set to zero, then it can be applied in a straight reach.

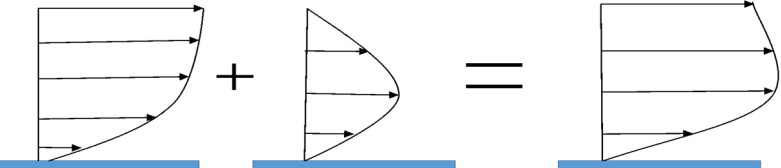

**Figure 2.** Synthesis of vertical velocity distribution in open-channel bends.

In this study, Equation (6) was slightly modified, and the following equation was derived:

$$u - \overline{u} = u' = \frac{u_*}{\kappa}(1 + \ln y') + a sin^2 \pi y' \tag{7}$$

where $a$ denotes the regression coefficient. In Equation (7), the logarithmic function proposed by van Karman is used and a sine function is added. The formula is similar to the "wake law" proposed by Coles [35]. This equation is advantageous because it can be quantitatively compared with Elder's results based on coefficient a when calculating the dispersion coefficient.

### 2.2. Derivation of Longitudinal Dispersion Coefficient

The two-dimensional longitudinal dispersion coefficient in open-channel bends can be obtained by substituting Equation (7) into Equation (2), followed by performing a triple integration. In the calculation, the vertical diffusion coefficient $\varepsilon$ is regarded as the depth-averaged coefficient $\overline{\varepsilon}$ without using a distribution formula such as Equation (3) to facilitate the integral calculation. The derivation procedure is included in Appendix A, and the result is:

$$\begin{aligned} D_L &= -\frac{d^2}{\overline{\varepsilon}}\left\{-0.0741\left(\frac{u_*}{\kappa}\right)^2 - 0.0196\frac{au_*}{\kappa} + 0.0258a^2\right\} \\ &= \frac{d^2}{\overline{\varepsilon}}\left\{-0.0258\left(a - 0.38\frac{u_*}{\kappa}\right)^2 + 0.0778\left(\frac{u_*}{\kappa}\right)^2\right\} \end{aligned} \tag{8}$$

In this equation, if $a = 0.38\frac{u_*}{\kappa}$, then the maximum value of the longitudinal dispersion coefficient becomes:

$$D_{L\ max} = 0.0778\left(\frac{d2}{\bar{\varepsilon}}\right)\left(\frac{u_*}{\kappa}\right)^2 \tag{9}$$

If the effect of curvature does not apply, i.e., $a = 0$, then the longitudinal dispersion coefficient becomes:

$$D_L = 0.0741\left(\frac{d2}{\bar{\varepsilon}}\right)\left(\frac{u_*}{\kappa}\right)^2 \tag{10}$$

Equation (10) should be identical to Elder's result (Equation (4)). A difference was observed between the two equations because Elder used a vertical distribution for the diffusion coefficient ($\varepsilon$) during triple integration, whereas in this study, we used $\varepsilon$ as the averaged value ($\bar{\varepsilon} = 0.067du_*$) to ease calculation. If we set the von Karman constant ($\kappa$) as 0.434 instead of 0.41, then the two equations are exactly the same. According to Rozovskii [36], because a von Karman constant of 0.5 has been suggested for open-channel bends, a slightly higher value can be adopted, as in this study.

### 2.3. Experiments in Two Open Channels

The proposed equations for the vertical velocity distribution and dispersion coefficient were compared and verified with the values measured from two different types of open channels, i.e., a small-scale laboratory channel and a mid-scale natural-like channel. The sinuosity of the laboratory channel in which the hydraulic experiment was performed was 1.32, and a schematic plan view is shown in Figure 3 [3]. As shown in this figure, the width of the channel was 1 m; meanwhile, alternating curvatures with a central angle of 120° at the center, as well as additional curvatures with a central angle of 60° at the inlet and outlet were present. The cross-section of the channel was rectangular, and a smooth bed was implemented using painted steel. Micro-acoustic Doppler velocimetry (ADV) was used to measure the flow structure; micro-ADV is a high-precision velocimetry method that measures a three-dimensional flow field. The flow velocity measurements were performed on 12 sections of the channel, as shown in Figure 3. Three cases (Cases 301, 303, and 402) of velocity field measurement were conducted based on the flow conditions in which the flow rate was varied from 0.03 to 0.09 m³/s [31]. Tracer tests were performed under the same flow conditions to elucidate the dispersion characteristics. A salt solution (NaCl) was used as the tracer.

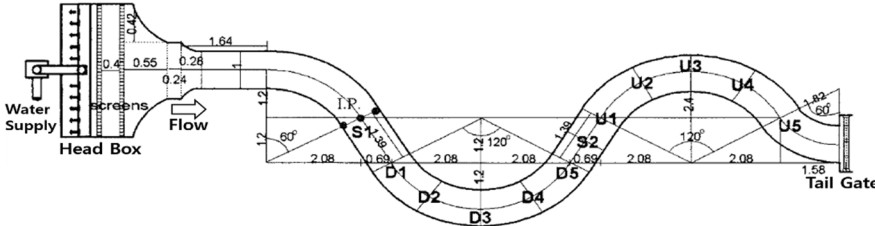

**Figure 3.** Plan view of small-scale meandering channel in laboratory (adapted from Baek et al. [3]).

A mid-scale nature-like channel was implemented at the Andong River Experiment Center, operated by KICT (Korea Institute of Construction Technology) in Korea. Three reaches with different sinuosity were connected in series, as shown in Figure 4 [37]. The bottom of the channel was composed of sand, and the cross-section of the channel exhibited a natural shape arising from the water flow. The length of the channel was 134 m for 1.5 sinuosity (A315 reach) and 155 m for 1.7 sinuosity (A317 reach). The three-dimensional flow field was measured using an acoustic Doppler current profiler at six sections of the channel, as shown in Figure 4. The average flow rate was 1.45 m³/s, and the average water depth was 0.487 m in the channel. The average cross-sectional velocity was measured to be 0.49–0.64 m/s in the A315 reach and 0.39–0.47 m/s in the A317 reach. A fluorescent

substance (rhodamine WT) was used as the tracer in the mixing experiment. The tracer was released at the injection point (shown in Figure 4) in each reach.

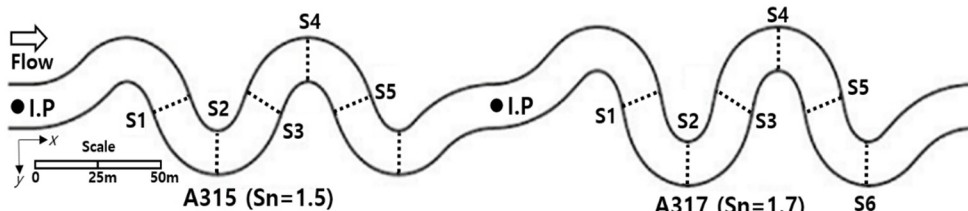

**Figure 4.** Plan view of mid-scale nature-like channel in Andong River Experiment Center in Korea (adapted from Shin et al. [37]).

## 3. Results and Discussion

### 3.1. Flow Characteristics in Meandering Channels

Before verifying the longitudinal dispersion coefficient proposed herein (Equation (8)), the vertical distribution equation for longitudinal velocity (Equation (7)) was fitted to the velocity data acquired from the meandering channels, and the regression coefficient a in Equation (7) was determined for all verticals. Figure 5 shows a comparison between the velocity obtained using the proposed equation and the observed velocity for each vertical section of Section U1 in the laboratory channel. As shown in this figure, in some cases, the flow velocity distribution equation agreed well with the observed value based on each vertical in one section. In particular, the flow velocity distribution was opposite to the logarithmic distribution (a distribution in which the flow velocity increases from the water surface to the bottom) in a certain vertical direction. In such cases, the velocity equation cannot appropriately describe the observed values. It is noteworthy that the point where the vertical distribution of the longitudinal velocity distorted significantly coincided with the point where the rotational cell of the secondary flow occurred.

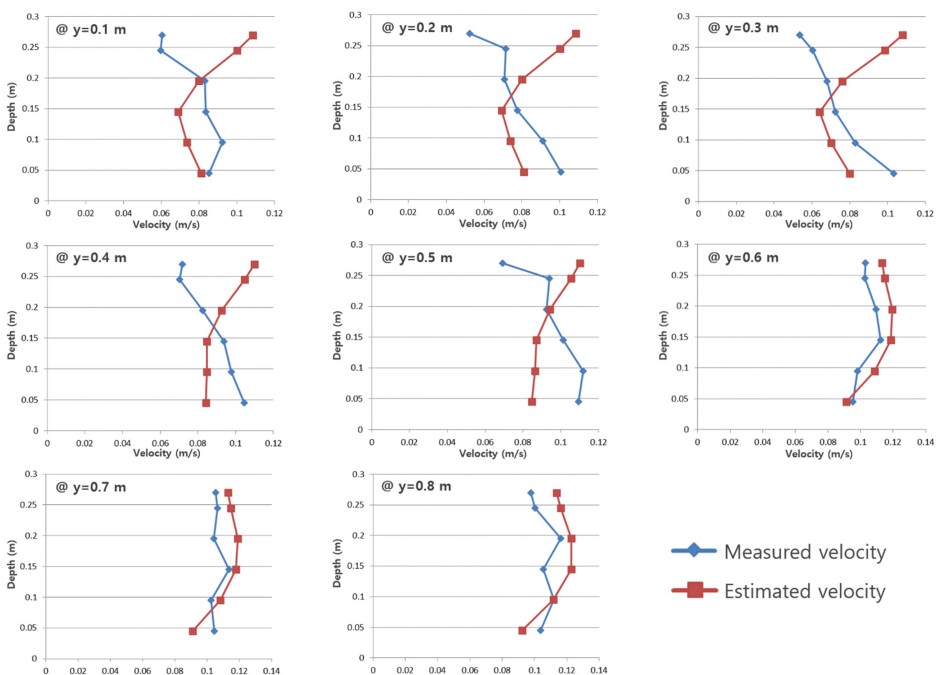

**Figure 5.** Comparison between measured velocity (blue line) and that estimated by Equation (7) (red line) in laboratory meandering channel at Section U1 (y denotes lateral coordinate from left to right bank).

To observe the structure of the secondary flow, the transverse and vertical velocity vectors occurring at the cross-sections were plotted, as shown in Figure 6A. In this figure, the evolution of the helical motion of the secondary flow along the channel is clearly illustrated. A typical two-cell system had developed weakly in Section D1 due to the effect of the entrance bend with a central angle of $60°$. After the first bend zone (Sections D3–D5), this two-cell helical motion disappeared, and a single large cell rotating in the clockwise direction appeared. This clockwise helical motion continued to develop until the entrance of the next alternating bend (Section U1). Proceeding to the apex of the bend, a cell that rotated in a counterclockwise direction began to develop at the lower section of the channel due to the centrifugal force (Section U3). The other cell, which occupied the upper section of the channel in Section U3, appeared to have prevented the development of the lower cell generated by the centrifugal force in the entire cross-section. This phenomenon is typically observed in curved channels with rectangular cross-sections and smooth roughness [38].

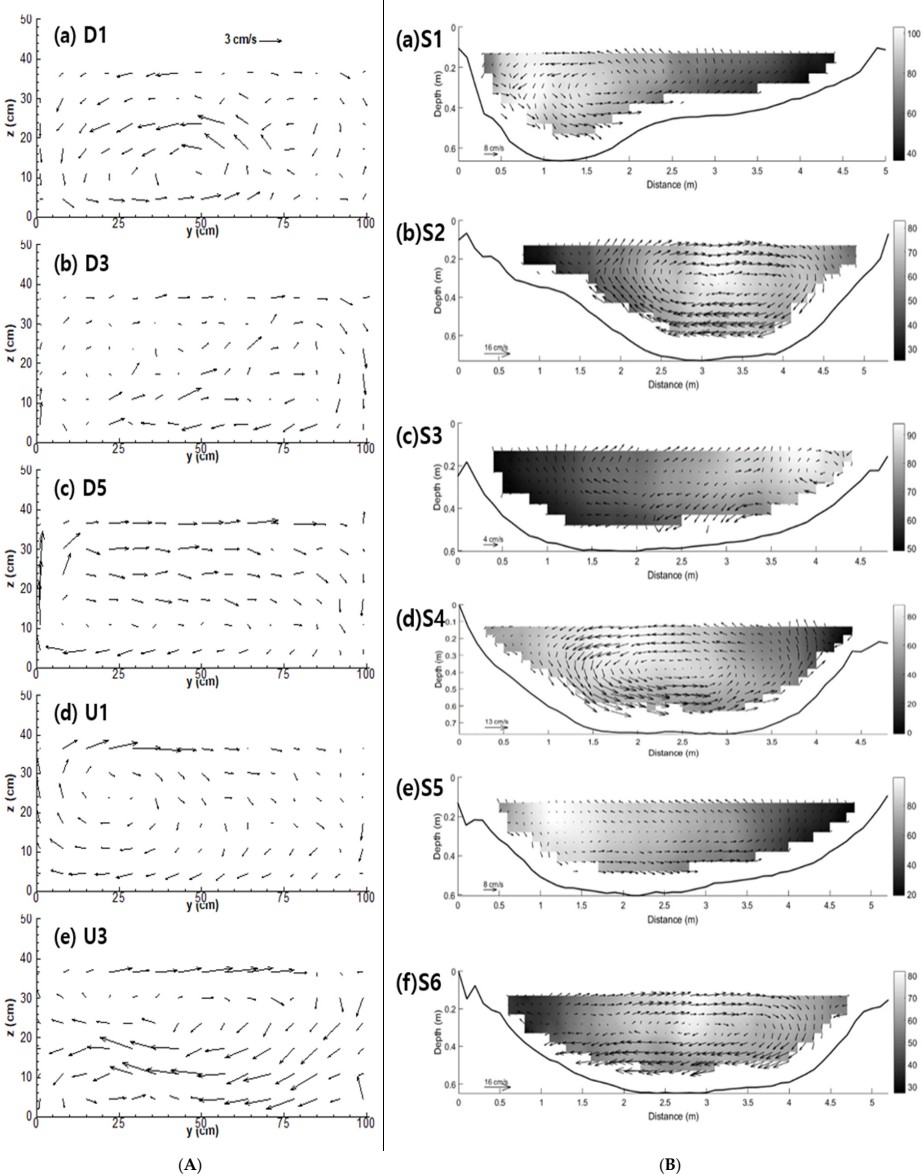

**Figure 6.** (**A**) Secondary flow patterns at representative cross-sections for case 402 in laboratory meandering channel (y denotes lateral coordinate from left to right bank, and z denotes vertical coordinate from bottom to water surface). (**B**) Secondary flow patterns at each cross-section of AMC 315 in nature-like channel.

As show in Figure 5, the vertical distribution of the longitudinal velocity near the left bank of Section U1 was severely distorted, such that the velocity near the bottom was almost twice that of the water surface. By contrast, the vertical distribution near the right bank exhibited a typical shape. It can be confirmed from Figure 6A that a strong rotating cell was generated on the left side of Section U1.

In the nature-like channel, as shown in Figure 6B, the pattern of the secondary flow differed from that in the laboratory channel. The typical helical motion behavior observed in streams with natural bed roughness and geometry can be observed in the nature-like channel. A large rotating cell with secondary flow occurred at the channel apexes of the even-numbered sections. The centrifugal force induced by the geometrical properties of the meandering channel transported the top of the water body to the outer bank at the channel apex. Clockwise and counterclockwise helical motion occurred alternatively in the bends. A strong cell was generated in the apex areas, whereas its strength decreased in the cross-over sections; as such, the growth and decay of the transverse flow was observed throughout the apex and cross-over areas in the channel.

Figures 7 and 8 show a comparison between the velocity obtained using the proposed equation and the measured velocity for each vertical section in the nature-like channel. By applying the proposed equation to the channel, the velocity distribution equation for each section was discovered to be relatively consistent with the measured values, and showed a higher applicability than the existing logarithmic formula.

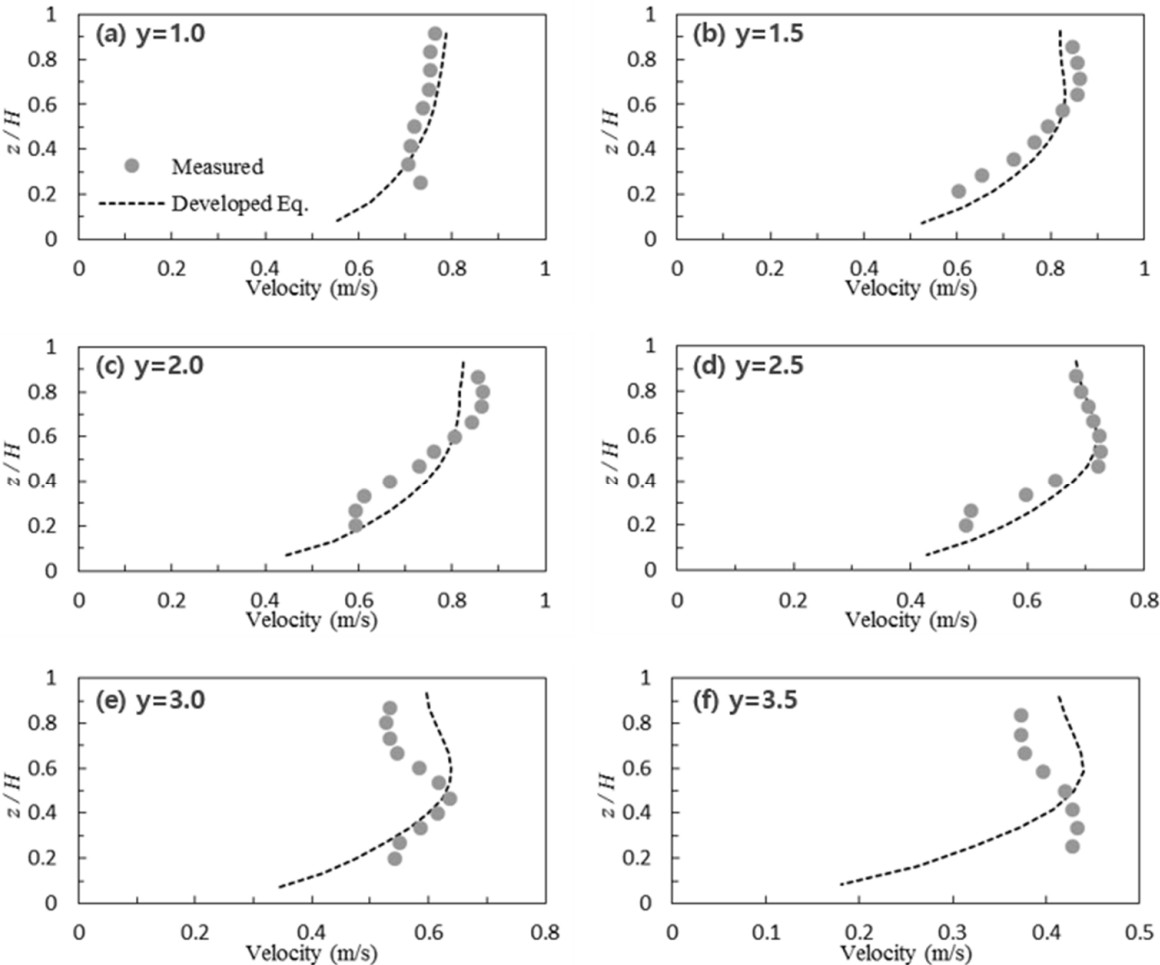

**Figure 7.** Comparison between measured velocity (dot) and that estimated by Equation (7) (line) at Section S4 of AMC 315 in nature-like channel (y denotes lateral coordinate from left to right bank).

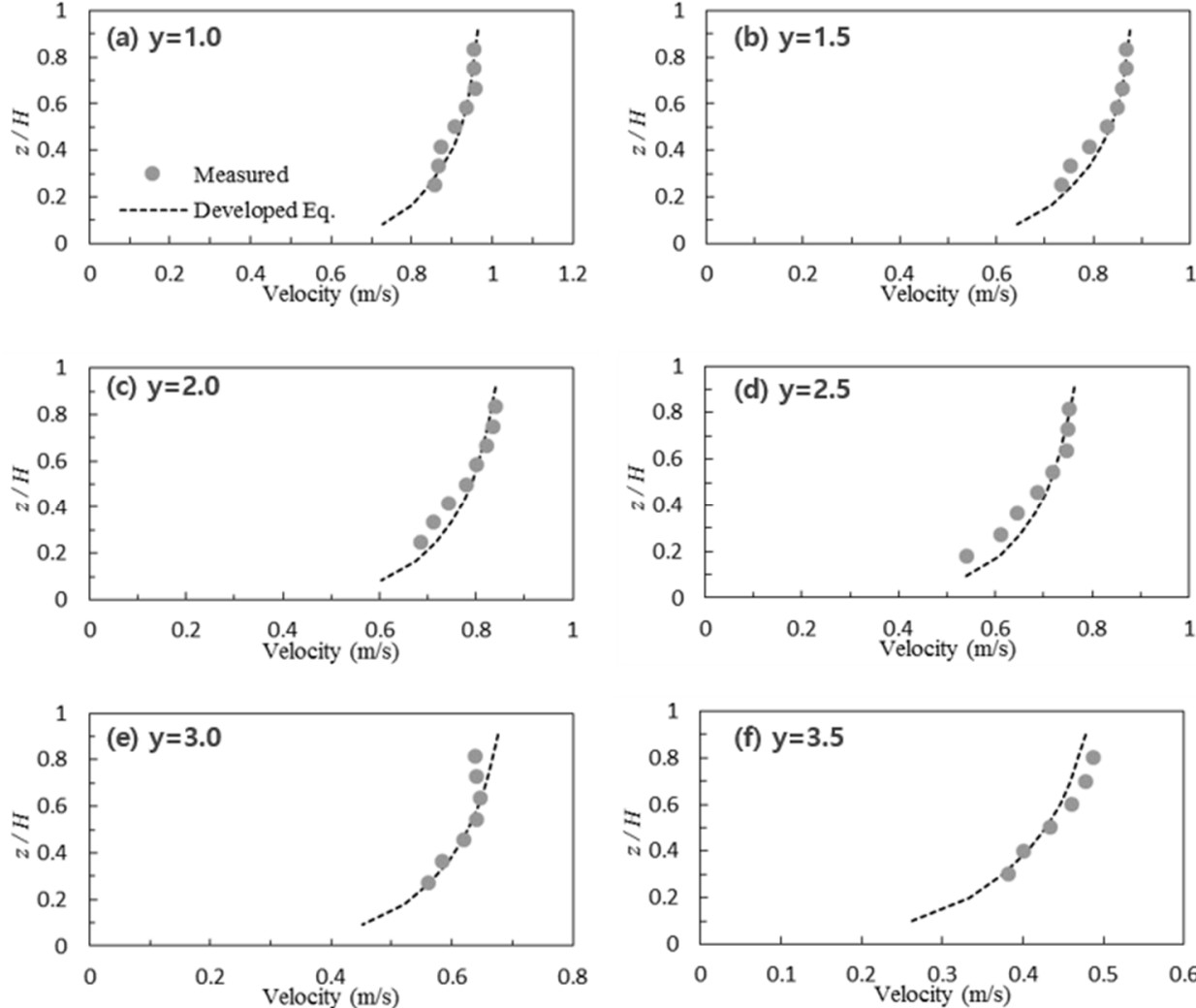

**Figure 8.** Comparison between measured velocity (dot) and that estimated by Equation (7) (line) at Section **S5** of AMC 315 in nature-like channel.

In the straight reach (odd-numbered sections), the observed velocity values and the values obtained using the proposed equation agreed well at all verticals (see Figure 8), whereas the values observed in the bends (even-numbered sections) differed from those of the logarithmic distribution at the verticals near the channel banks (see Figure 7). Consequently, in the section where strong rotational flow occurred in the nature-like channel, the flow velocity distribution distorted significantly, which is consistent with the experimental results of the laboratory meandering channel.

### 3.2. Longitudinal Dispersion Coefficient in Meandering Channels

The longitudinal dispersion coefficient of the proposed equation (Equation (8)) was compared with the observed dispersion coefficient acquired from the two-dimensional routing procedure, which was based on the concentration data obtained from tracer experiments in the laboratory channel. Baek et al. [3] developed a two-dimensional routing procedure and calculated the longitudinal and transverse dispersion coefficients for identical cases in a laboratory channel. The comparison results are presented in Table 1. In this table, the value of *a* in Equation (7) when the longitudinal dispersion coefficient has the maximum value, and the corresponding maximum value of the dimensionless coefficient ($D_L/du_*$) are included as well.

**Table 1.** Comparison between observed dispersion coefficients and estimated ones for laboratory and nature-like channels.

| Channel | Case Number | $a$ (Maximum Value) | $D_L/du_*$ | | Observed Value |
|---|---|---|---|---|---|
| | | | This Study (Maximum Value) | Adapted from Elder [5] ($a = 0$) (Minimum Value) | |
| Lab. | 301 | 0.0059 | 6.22 | 5.93 | 9.2 |
| | 303 | 0.0178 | 6.22 | | 4.5 |
| | 402 | 0.0092 | 6.22 | | 4.3 |
| Nature-like | A315 | 0.092 | 6.89 | | 4.72~8.58 |
| | A317 | 0.103 | 6.90 | | 5.38~9.82 |

The observed coefficient based on laboratory concentration data and the coefficient calculated using the formula proposed herein did not differ significantly. However, the fact that the velocity equation proposed in this study does not appropriately describe the velocity distribution in many verticals and that the longitudinal dispersion coefficients in natural rivers are much larger than the values presented in the laboratory channel suggest the necessity to perform a comparison with data obtained from the nature-like channel.

The comparison results for the nature-like channels are summarized in Table 1. The observed dispersion coefficient was obtained using the two-dimensional stream-tube routing procedure [39], which can reflect the irregularities of the channel's geometry. The dispersion coefficient calculated using Equation (8), and the observed dispersion based on the concentration data, showed relatively similar results, although the observation-based value range was higher.

In terms of the formula proposed herein, the range of the calculated dispersion coefficient is limited because it is a transformation formula based on a logarithmic velocity distribution. The maximum value of the dimensionless dispersion coefficient ($D_L/du_*$) obtained using the formula is 6.90, whereas the observed value is approximately 10.

## 4. Conclusions

In this study, an equation that can appropriately describe the vertical distribution of flow velocity in an open-channel bend was proposed, and a two-dimensional longitudinal dispersion coefficient was theoretically derived based on the velocity equation. The proposed velocity equation and dispersion coefficient were verified using two datasets acquired from a laboratory curve channel and a nature-like channel.

In terms of flow velocity, depending on each vertical in one section, the flow velocity distribution equation proposed herein agreed well with the measured values in only some cases. In particular, the flow velocity distribution was opposite to the logarithmic distribution (a distribution in which the flow velocity increases from the water surface to the bottom). In this case, the velocity equation did not describe the observed values appropriately. The continuous increase in flow velocity toward the bottom not only occurred in the laboratory artificial channel, but also in natural rivers. It is noteworthy that the point where the vertical distribution of the longitudinal flow velocity was severely distorted coincided with the point where the secondary rotational cell occurred. In terms of the longitudinal dispersion coefficient, no significant difference was observed between the observed dispersion coefficient based on the concentration data and the coefficient value calculated using the equation proposed herein.

The range of the calculated dispersion coefficient was limited because it is a transformation formula based on a logarithmic distribution of the velocity distribution. Therefore, the equation for dispersion proposed herein is reasonable for the longitudinal dispersion coefficient of bends of an open channel. However, the fact that the velocity distribution equation proposed herein did not appropriately describe the velocity distribution in some

verticals should be investigated more comprehensively in the future. In other words, a distribution formula that can describe a severely distorted flow velocity distribution, such as an inverted log distribution, should be identified in the future.

**Author Contributions:** Conceptualization, K.O.B. and W.S.II; methodology, K.O.B.; model simulation, K.O.B.; validation, W.S.II; formal analysis, K.O.B. and W.S.II; writing—review and editing, K.O.B. and W.S.II. All authors have read and agreed to the published version of the manuscript.

**Funding:** This research was funded by K-water (20-B-W-014), National Research Foundation of Korea (NRF-2016R1D1A1B02012110), and Korea Ministry of Environment (MOE) (2021003110003).

**Institutional Review Board Statement:** Not applicable.

**Informed Consent Statement:** Not applicable.

**Data Availability Statement:** Not applicable.

**Acknowledgments:** This study was supported by K-water (20-B-W-014) and by the National Research Foundation of Korea (NRF-2016R1D1A1B02012110). Also this study was partially supported by Korea Ministry of Environment (MOE) (2021003110003).

**Conflicts of Interest:** The authors declare no conflict of interest.

## Appendix A

Derivation of longitudinal dispersion coefficient.

$$D_L = -\frac{1}{d}\int_0^d u' \int_0^y \frac{1}{\varepsilon}\int_0^y u'\,dy\,dy\,dy$$

Let

1. $y' = \frac{y}{d}$
2. $u' = \frac{u_*}{\kappa}(1+\ln y') + a\sin^2\pi y'$
3. $\varepsilon$ : constant

$\int_0^y u'\,dy = \int_0^y \left\{\frac{u_*}{\kappa}(1+\ln y') + a\sin^2\pi y'\right\}dy \dots (1)$

$(1) = d\int_0^{y'}\left\{\frac{u_*}{\kappa}(1+\ln y') + a\sin^2\pi y'\right\}dy' = d\int_0^{y'}\left\{\frac{u_*}{\kappa}(1+\ln y') + \frac{a}{2}(1-\cos2\pi y')\right\}dy'$

$= d\left\{\frac{u_*}{\kappa}y'\ln y'\Big|_0^{y'} + \frac{a}{2}\left(y' - \frac{1}{2\pi}\sin2\pi\right)\Big|_0^{y'}\right\} = d\left\{\frac{u_*}{\kappa}y'\ln y' + \frac{a}{2}\left(y' - \frac{1}{2\pi}\sin2\pi\right)\right\}$

$\int_0^y \frac{1}{\varepsilon}\int_0^y u'\,dy\,dy = \frac{d}{\varepsilon}\int_0^y\left\{\frac{u_*}{\kappa}y'\ln y' + \frac{a}{2}\left(y' - \frac{1}{2\pi}\sin2\pi\right)\right\}dy = \frac{d^2}{\varepsilon}\int_0^{y'}\left\{\frac{u_*}{\kappa}y'\ln y' + \frac{a}{2}\left(y' - \frac{1}{2\pi}\sin2\pi\right)\right\}dy' \dots (2)$

$(2) = \frac{d^2}{\varepsilon}\left\{\frac{u_*}{\kappa}\left(\frac{1}{2}y'^2\ln y' - \frac{1}{4}y'^2\right)\Big|_0^{y'} + \frac{a}{2}\left(\frac{1}{2}y'^2 + \frac{1}{4\pi^2}\cos2\pi y'\right)\Big|_0^{y'}\right\}$

$= \frac{d^2}{\varepsilon}\left\{\frac{u_*}{\kappa}\left(\frac{1}{2}y'^2\ln y' - \frac{1}{4}y'^2\right) + a\left(\frac{1}{4}y'^2 + \frac{1}{8\pi^2}(\cos2\pi y' - 1)\right)\right\}$

$-\frac{1}{d}\int_0^d u'\int_0^y\frac{1}{\varepsilon}\int_0^y u'\,dy\,dy\,dy$

$= -\frac{1}{d}\times\frac{d^2}{\varepsilon}\int_0^d\left\{\frac{u_*}{\kappa}(1+\ln y') + \frac{a}{2}(1-\cos2\pi y')\right\}\begin{Bmatrix}\frac{u_*}{\kappa}\left(\frac{1}{2}y'^2\ln y' - \frac{1}{4}y'^2\right)+ \\ a\left(\frac{1}{4}y'^2 + \frac{1}{8\pi^2}(\cos2\pi y' - 1)\right)\end{Bmatrix}dy \dots (3)$

$(3) = -\frac{d^2}{\varepsilon}\int_0^1\left\{\frac{u_*}{\kappa}(1+\ln y') + \frac{a}{2}(1-\cos2\pi y')\right\}\begin{Bmatrix}\frac{u_*}{\kappa}\left(\frac{1}{2}y'^2\ln y' - \frac{1}{4}y'^2\right)+ \\ a\left(\frac{1}{4}y'^2 + \frac{1}{8\pi^2}(\cos2\pi y' - 1)\right)\end{Bmatrix}dy'$

$= -\frac{d^2}{\varepsilon}\left\{\int_0^1\left(\frac{u_*}{\kappa}\right)^2(1+\ln y')\left(\frac{1}{2}y'^2\ln y' - \frac{1}{4}y'^2\right)dy'\right.$

$\qquad + \int_0^1\frac{au_*}{\kappa}(1+\ln y')\left(\frac{1}{4}y'^2 + \frac{1}{8\pi^2}(\cos2\pi y' - 1)\right)dy'$

$\qquad + \int_0^1\frac{au_*}{\kappa}\left(\frac{1-\cos2\pi y'}{2}\right)\left(\frac{1}{2}y'^2\ln y' - \frac{1}{4}y'^2\right)dy'$

$\qquad \left. + \int_0^1 a^2\left(\frac{1-\cos2\pi y'}{2}\right)\left(\frac{1}{4}y'^2 + \frac{1}{8\pi^2}(\cos2\pi y' - 1)\right)dy'\right\} \dots (4)$

Set

$A = \int_0^1\left(\frac{u_*}{\kappa}\right)^2(1+\ln y')\left(\frac{1}{2}y'^2\ln y' - \frac{1}{4}y'^2\right)dy'$

$B = \int_0^1\frac{au_*}{\kappa}(1+\ln y')\left(\frac{1}{4}y'^2 + \frac{1}{8\pi^2}(\cos2\pi y' - 1)\right)dy'$

$C = \int_0^1\frac{au_*}{\kappa}\left(\frac{1-\cos2\pi y'}{2}\right)\left(\frac{1}{2}y'^2\ln y' - \frac{1}{4}y'^2\right)dy'$

$D = \int_0^1 a^2\left(\frac{1-\cos2\pi y'}{2}\right)\left(\frac{1}{4}y'^2 + \frac{1}{8\pi^2}(\cos2\pi y' - 1)\right)dy'$

Then

$$(4) = -\frac{d^2}{\varepsilon}\{A + B + C + D\}$$

In here

$$A = \int_0^1 \left(\frac{u_*}{\kappa}\right)^2 (1 + \ln y')\left(\frac{1}{2}y'^2 \ln y' - \frac{1}{4}y'^2\right)dy'$$

$$= \left(\frac{u_*}{\kappa}\right)^2 \left\{ (y' \ln y') \times \left(\frac{1}{2}y'^2 \ln y' - \frac{1}{4}y'^2\right)\Big|_0^1 - \int_0^1 (y' \ln y')^2 dy' \right\}$$

$$= -\left(\frac{u_*}{\kappa}\right)^2 \int_0^1 (y' \ln y')^2 dy' - \left(\frac{u_*}{\kappa}\right)^2 \left\{ \left(\frac{1}{3}y'^3(\ln y')^2\right)\Big|_0^1 - \int_0^1 \frac{1}{3}y'^3 \times 2\ln y' \times \frac{1}{y'}dy' \right\}$$

$$= \left(\frac{u_*}{\kappa}\right)^2 \times \frac{2}{3} \times \int_0^1 y'^2 \ln y' dy'$$

$$= \left(\frac{u_*}{\kappa}\right)^2 \times \frac{2}{3} \times \left(\frac{-1}{3^2}\right) = -\frac{2}{27}\left(\frac{u_*}{\kappa}\right)^2 = -0.0741\left(\frac{u_*}{\kappa}\right)^2$$

$$B = \int_0^1 \frac{au_*}{\kappa}(1 + \ln y')\left(\frac{1}{4}y'^2 + \frac{1}{8\pi^2}(cos2\pi y' - 1)\right)dy'$$

$$= \frac{au_*}{\kappa}\left\{ (1 + \ln y')\left(\frac{1}{12}y'^3 + \frac{1}{8\pi^2}\left(\frac{1}{2\pi}sin2\pi y' - y'\right)\right)\Big|_0^1 - \int_0^1 \frac{1}{y'}\left(\frac{1}{12}y'^3 + \frac{1}{8\pi^2}\left(\frac{1}{2\pi}sin2\pi y' - y'\right)\right)dy' \right\}$$

$$= \frac{au_*}{\kappa}\left\{ \left(\frac{1}{12} - \frac{1}{8\pi^2}\right) - \int_0^1 \left(\frac{1}{12}y'^2 + \frac{1}{8\pi^2}\left(\frac{1}{2\pi y'}sin2\pi y' - 1\right)\right)dy' \right\}$$

$$= \frac{au_*}{\kappa}\left\{ \frac{1}{18} - \int_0^1 \frac{sin2\pi y'}{16\pi^3 y'}dy' \right\} = \frac{au_*}{\kappa}\left\{ \frac{1}{18} - \frac{1}{16\pi^3}\int_0^1 \frac{sin2\pi y'}{y'}dy' \right\}$$

$$= \frac{au_*}{\kappa}\left\{ \frac{1}{18} - \frac{1}{16\pi^3}\sum_{n=1}^{\infty}\frac{(2\pi)^{2n-1}(-1)^{n-1}}{(2n-1)!(2n-1)} \right\} = \frac{0.0527au_*}{\kappa}$$

$$C = \int_0^1 \frac{au_*}{\kappa}\left(\frac{1 - cos2\pi y'}{2}\right)\left(\frac{1}{2}y'^2 \ln y' - \frac{1}{4}y'^2\right)dy'$$

$$= \frac{au_*}{\kappa}\left[ \left(\frac{1}{2}y' - \frac{sin2\pi y'}{4\pi}\right)\left(\frac{1}{2}y'^2 \ln y' - \frac{1}{4}y'^2\right)\Big|_0^1 - \int_0^1 \left(\frac{1}{2}y' - \frac{sin2\pi y'}{4\pi}\right)(y' \ln y')dy' \right]$$

$$= \frac{au_*}{\kappa}\left[ -\frac{1}{8} - \int_0^1 \frac{1}{2}y'^2 \ln y' dy' + \frac{1}{4\pi}\int_0^1 sin2\pi y'(y' \ln y')dy' \right]$$

$$= \frac{au_*}{\kappa}\left[ -\frac{1}{8} - \frac{1}{2} \times \left(\frac{-1}{3^2}\right) + \frac{1}{4\pi}\int_0^1 sin2\pi y'(y' \ln y')dy' \right]$$

$$= \frac{au_*}{\kappa}\left[ -\frac{5}{72} + \frac{1}{4\pi} \times \sum_{n=1}^{\infty}\frac{(2\pi)^{2n-1}(-1)^n}{(2n-1)!(2n+1)^2} \right] = -0.0723\frac{au_*}{\kappa}$$

$$D = \int_0^1 a^2\left(\frac{1 - cos2\pi y'}{2}\right)\left(\frac{1}{4}y'^2 + \frac{1}{8\pi^2}(cos2\pi y' - 1)\right)dy'$$

$$= \frac{a^2}{2}\int_0^1 (1 - cos2\pi y')\left(\frac{1}{4}y'^2 - \frac{1}{8\pi^2}(1 - cos2\pi y')\right)dy'$$

$$= \frac{a^2}{2}\left[\int_0^1 \frac{1}{4}y'^2(1 - cos2\pi y')dy' - \frac{1}{8\pi^2}\int_0^1 (1 - cos2\pi y')^2 dy'\right]$$

$$= \frac{a^2}{2}\left[\int_0^1 \frac{1}{4}y'^2 dy' - \frac{1}{4}\int_0^1 y'^2 cos2\pi y' dy' - \frac{1}{8\pi^2}\int_0^1 \left(\frac{3}{2} - 2cos2\pi y' + \frac{1}{2}cos4\pi y'\right)dy'\right]$$

$$= \frac{a^2}{2}\left[\frac{1}{12} - \frac{1}{4}\int_0^1 y'^2 cos2\pi y' dy' - \frac{3}{16\pi^2}\right]$$

$$= \frac{a^2}{2}\left[\frac{1}{12} - \frac{1}{8\pi^2} - \frac{3}{16\pi^2}\right] = \frac{a^2}{2}\left[\frac{1}{12} - \frac{5}{16\pi^2}\right] = 0.0258a^2$$

Finally,

$$D_L = -\frac{d^2}{\varepsilon}\{A + B + C + D\}$$

$$= -\frac{d^2}{\varepsilon}\left\{ -0.0741\left(\frac{u_*}{\kappa}\right)^2 + 0.0527\frac{au_*}{\kappa} - 0.0723\frac{au_*}{\kappa} + 0.0258a^2 \right\}$$

$$= \frac{d^2}{\varepsilon}\left\{ -0.0258\left(a - 0.38\frac{u_*}{\kappa}\right)^2 + 0.0778\left(\frac{u_*}{\kappa}\right)^2 \right\}$$

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
