# Peer review of "Modifying Elder’s Longitudinal Dispersion Coefficient for Two-Dimensional Solute Mixing Analysis in Open-Channel Bends"

_water, doi:10.3390/w14192962_

Round 1

Reviewer 1 Report

See attached file.

Reviewer 2 Report

Dear Prof. Baek / Prof. Seo;

You are leader in river mixing studies and I'm fully familiar with your valuable contributions to environmental hydraulic discipline. Frankly, I enjoyed reading the paper and believe it is fully within the aims and scope of Water. Therefore, I would suggest acceptance with considering minor suggestions as given below:

1- Abstract can end with some implications of the findings in a broader context. For example, what can others learn from your investigation? How can they apply your findings to their own case studies?

2- It would be better to avoid using the Title words in Keywords.

3- Figures 3 and 4: I think you need a permission for both figures (I'm not sure). Please discuss this matter with the journal publication team.

Good luck

Roohollah Noori

Author Response

The authors appreciate the reviewer’s detailed comments.

Herein, we provide responses and the details of the revisions to the manuscript, point by point.

1- Abstract can end with some implications of the findings in a broader context. For example, what can others learn from your investigation? How can they apply your findings to their own case studies?

Response> According to the reviewer’s comment, the abstract was revised.

2- It would be better to avoid using the Title words in Keywords.

Response> According to the reviewer’s comment, the keywords were revised.

3- Figures 3 and 4: I think you need a permission for both figures (I'm not sure). Please discuss this matter with the journal publication team.

Response> The authors appreciate your detailed advice. We will discuss this matter with the journal publication team in the event the manuscript is accepted for publication.

Round 2

Reviewer 1 Report

After reading the attached documents, namely improved manuscript, I believe that all my suggestions were sufficiently addessed and manuscript is ready for publishing in WATER.